# Boosting Self-Supervised Graph Representation Learning via Anchor-Neighborhood Alignment and Isotropic Constraints

## Abstract

Graph Self-Supervised Learning (GSSL) provides a guarantee for harnessing abundant unlabeled data and has attracted widespread attention. While making some strides, GSSL still faces several crucial challenges that hinder it from fully unleashing its potential, including inadequate exploration of graph information and underlying collapse issues. To overcome these obstacles, we propose two complementary components aimed at sufficiently mining valuable contents implied within graphs and transforming them into informative and diverse representations through training an expressive neural model. As a cornerstone module, an anchor-neighborhood alignment strategy, which utilizes graph diffusion to construct the probability distribution of positive samples based on the structural context of the anchor node, enables sufficient exploration of graph topology and endows the neural model with stronger structure-aware ability. To enhance diversity of node representations, a scheme of isotropic constraints is introduced to encourage representations to exhibit consistent distribution along any direction in space, which compels data points to be scattered throughout the whole representation space and naturally solves the notorious dimensional collapse in self-supervised learning. Owing to no reliance on negative samples, mutual information estimators, and additional projectors, our approach presents significant advantages in computing and storage. Extensive comparative experiments and exhaustive ablation studies demonstrate the effectiveness and efficiency of our method.

## 1 Introduction

Graph-structured data is ubiquitous in real-world scenarios due to its powerful capability to model relational entities. Representative examples include citation networks (Sen et al., 2008), social networks (Yanardag & Vishwanathan, 2015), and knowledge graphs (Vivona & Hassani, 2019). As an effective tool for processing graph data, Graph Neural Networks (GNNs) (Kipf & Welling, 2016a; Veličković et al., 2017) have drawn widespread attention and made significant strides. Most GNN-based models are established in a supervised mode, which typically necessitates plentiful labeled data for training. Nevertheless, annotating graphs is a challenging mission, and task-related labels would be extremely scarce. Hence, learning high-quality graph representations without manual annotations for various downstream tasks constitutes a pivotal subject within the domain of graph machine learning. Among state-of-the-art unsupervised methods, multi-view graph Self-Supervised Learning (SSL) has achieved promising performance and even surpasses their supervised counterparts. These methods typically follow the technical route of multi-view learning, aiming to produce invariant representations across distinct augmented views (*i.e.*, positive sample pairs) by maximizing their agreement. Nevertheless, a solo pursuit of this objective will lead to a completely collapsed solution where all node representations approximately shrink to a constant vector, as shown in Figure 1(a). The existing graph self-supervised learning methods usually prevent the training from falling into this dilemma by pushing away the embedding vectors of negative sample pairs (Chen et al., 2020; Zhu et al., 2020; 2021), employing an asymmetric architecture (Grill et al., 2020; Thakoor et al., 2021), or decorrelating various representation channels (Zhang et al., 2021).

Despite the ascendance and commendable performance of the multi-view paradigm of graph SSL, these methods still suffer some notable drawbacks, impeding further improvements in their performance and efficiency. Without reliance on manual annotations, self-supervised learning excavates

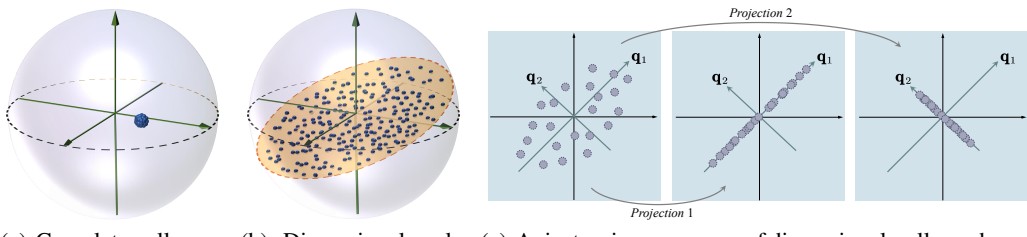

(a) Complete collapse.    (b) Dimensional collapse.    (c) Anisotropic appearance of dimensional-collapsed representations.

Figure 1: Illustration of collapse issues and their distinct manifestations.

supervisory signals, extracts valuable knowledge, and trains expressive models via thoughtfully crafted pretext tasks (Liu et al., 2023). Generally speaking, a more comprehensive information retrieval from unlabeled data usually results in enhanced performance of self-supervised learning. Especially, topological structure, which distinguishes graphs from regular grid data like images, provides abundant structural priors and ample self-supervisory signals. However, current methods struggle to harness this rich topological information adequately. Typically, when constructing positive samples, a crucial component of multi-view self-supervised learning, these methods often straightforwardly consider various augmented versions of the same instance. This practice neglects topological priors within graphs, inevitably leading to suboptimal performance. Although some pioneers (Peng et al., 2020; Mo et al., 2022) adopted neighbors of the anchor node as positive samples, the sampled range is only limited to one-hop in most cases, and all sampled nodes are treated equally. Besides, graph SSL acquires knowledge from a vast pool of unlabeled data, which inherently necessitates neural models with strong expressiveness to encapsulate extracted structural information and semantic content effectively. Nevertheless, the knotty issues of mode collapse in SSL significantly hamper the expressive capacity of models and the diversity of representations. In addition to the apparent complete collapse, another potential concern is dimensional collapse (Hua et al., 2021) in Figure 1(b), where the representations converge into a subspace and various channels are tightly coupled together. This issue leads to uninformative representations and inexpressive models. Lastly, and by no means least, it is vital to underscore the significance of training efficiency and its associated costs during self-supervised pre-training. For most preceding approaches, apart from feature extractors, additional designs and architectures are still required, such as projection heads (You et al., 2020; Zhu et al., 2021; 2020) and parameterized mutual information estimators (Hassani & Khasahmadi, 2020; Sun et al., 2020), which add extra storage and computing burden.

Motivated by the aforementioned analysis, this research strives to construct a comprehensive graph self-supervised learning framework to thoroughly mine information from unlabeled data and enhance the diversity of representations in an efficient manner. Deviating from most pioneer works, we refrain from directly pairing two augmented views of one instance (i.e., anchor node) as positive samples. Apart from data augmentation, graph diffusion is employed to establish the distribution of positive samples based on structural context of the anchor node, sufficiently capturing topological relationships within the graph. Subsequently, the positive samples with diffusion weights are derived from the constructed distribution to realize weighted anchor-neighborhood alignment in representation space to endow the model with stronger structure-aware ability. Furthermore, it is imperative to eliminate collapse issues to enhance the expressiveness of models for accommodating extracted knowledge. Beyond the prevalent notion of dimensional collapse, characterized by high inter-channel correlations, another underlying phenomenon has not received explicit attention, where representations display divergent distributions along distinct directions (referred to as anisotropy), as depicted in Figure 1(c). In light of our observations, we introduce an approach of isotropic constraints to mitigate dimensional collapse, encouraging representations to exhibit a consistent distribution across all spatial directions and achieve diversity. Our research advances the study of dimensional collapse in SSL and provides a unified explanation for distinct self-supervised methods from the standpoint of optimization objectives and final outcomes. Combining Anchor-Neighborhood Alignment with Isotropic Constraints (ANA-IC), this paper builds a thorough graph SSL framework with two complementary components. Due to no reliance on additional architectures such as mutual information estimators and projection heads, the lightweight design dramatically enhances the efficiency of our model. To sum up, our contributions are as follows:

- A strategy of anchor-neighborhood alignment is put forward to enhance the comprehensive exploration of unlabeled graph data and reinforce topological awareness of neural models, whose core is an ingenious positive sampling scheme based on structural context distribution.

- We reexamine another manifestation of dimension collapse, namely, anisotropic distributions. Correspondingly, an innovative scheme of isotropic constraints is proposed to mitigate collapse issues and enhance representation diversity, independent of existing methods such as negative sampling, channel decorrelation, and asymmetric architectures.

- Extensive experiments and analysis demonstrate the effectiveness and efficiency of our approach when juxtaposed with state-of-the-art baselines. Besides, exhaustive ablation studies and visual analysis provide deeper insights into the underlying principles and advantages of our methodology.

## 2 RELATED WORK

### 2.1 MULTI-VIEW GRAPH SELF-SUPERVISED LEARNING

Recently, numerous research efforts have been devoted to graph self-supervised learning, and a branch based on multi-view learning stands out due to its superior performance. The basic idea is to make multiple views from the same instance under various graph transformations agree with each other to optimize model parameters. Despite the similarity in principle, these methods are intricately crafted and diverge in multiple aspects, such as network architecture, view design, and self-supervised pretext tasks. Enlighted by the InfoMax principle (Linsker, 1988), Deep Graph Infomax (DGI) (Veličković et al., 2018) and InfoGraph (Sun et al., 2020) focus on maximizing mutual information between patch-level representations and a graph-level summary vector using a parameterized Jenson-Shannon estimator (Nowozin et al., 2016) for graph representation learning. Incorporating domain-specific priors, GraphCL (You et al., 2020) systematically investigates the impact of various combinations of graph transformations for classification tasks. MVGRL (Hassani & Khasahmadi, 2020) employs graph diffusion (Klicpera et al., 2019) to construct augmented views and performs cross-view contrastiveness between local and global representations. GGD (ZHENG et al., 2022) revisits the underlying principles of DGI and MVGRL and puts forward an efficient group discrimination method for graph contrastive learning. GRACE (Zhu et al., 2020) and GCA (Zhu et al., 2021) utilize an improved InfoNCE (Gutmann & Hyvärinen, 2010) loss as their objective function, where multiple views are generated by edge perturbation and attribute masking. As an enhancement, GCA employs node centrality to perform adaptive augmentations. G-BT (Bielak et al., 2022) generalizes the well-known Barlow Twins (Zbontar et al., 2021) in computer vision onto graph field. CCA-SSG (Zhang et al., 2021) introduces canonical correlation analysis to graph self-supervised learning, which discards negative samples and shifts towards inter-channel repulsion.

### 2.2 TACKLING COLLAPSE ISSUES IN SELF-SUPERVISED LEARNING

Preventing collapse is a fundamental concern in self-supervised learning, with the primary forms of collapse encompassing complete collapse and dimensional collapse. Some methods, such as MOCO (He et al., 2020), SimCLR (Chen et al., 2020), GRACE (Zhu et al., 2020), and GCA (Zhu et al., 2021), utilize negative samples to push representations of different instances away from each other, which is equivalent to realizing mutual information maximization. Another line of approaches adopts asymmetric architecture and stop-gradient strategy to prevent all representations from shrinking to the same point and avoid complete collapse. Representative examples include BYOL (Grill et al., 2020), SimSiam (Chen & He, 2021), and BGRL (Thakoor et al., 2021). Moreover, several additional methods, such as VICReg (Bardes et al., 2022), CCA-SSG (Zhang et al., 2021), and Barlow Twins (Zbontar et al., 2021), utilize feature decorrelation to alleviate collapse issues by decoupling various channels. Diverging from previous research, our approach enforces isotropic constraints on representations, which provides a natural solution to address collapse issues.

## 3 METHODOLOGY

In this section, the method is expanded in a progressive manner: commencing with introducing relevant notions and the basic framework, followed by the elucidation of the anchor-neighborhood alignment strategy, and demonstrating the principle and implementation of isotropic constraints.

### 3.1 PRELIMINARIES AND BASIC FRAMEWORK

**Notations.** A graph with $N$ nodes, denoted as $G(\mathbf{A}, \mathbf{X})$, is characterized by its node set $\mathcal{V} = \{v_1, ..., v_N\}$ and edge set $\mathcal{E}$. Each node $v_i \in \mathcal{V}$ possesses a $D$-dimensional feature vector $\mathbf{x}_i \in \mathbb{R}^D$.

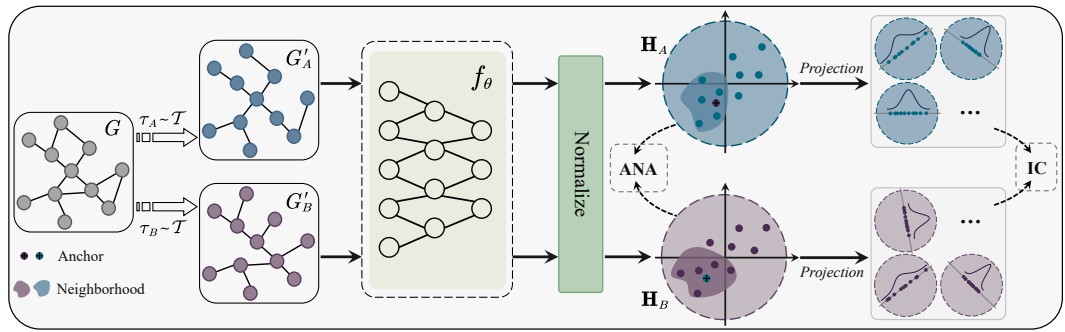

Figure 2:    The basic framework of the proposed ANA-IC. The "Neighborhood" covers structural neighbors. Best viewed in colors.

Node feature matrix $\mathbf{X} = [\mathbf{x}_1, ..., \mathbf{x}_N]^\top \in \mathbb{R}^{N \times D}$ covers attribute information for all nodes while adjacency matrix $\mathbf{A} \in \mathbb{R}^{N \times N}$ characterize topological connections within the graph. The objective of graph self-supervised learning is to learn a continuous mapping $f_\theta(\mathbf{A}, \mathbf{X}) : \mathbb{R}^{N \times N} \times \mathbb{R}^{N \times D} \to \mathbb{R}^{N \times d}$ with learnable parameters $\theta$ and representation dimension $d$ through carefully constructed pretext tasks, which can output generalized representations $\widetilde{\mathbf{H}} = [\tilde{\mathbf{h}}_1, ..., \tilde{\mathbf{h}}_N]^\top \in \mathbb{R}^{N \times d}$ for various downstream tasks. For any matrix $\mathbf{M}$, $M_{ij}$ indicates the element in the $i$-th row and $j$-th column.

**View Generation.**    An augmented version $G'(\mathbf{A}', \mathbf{X}')$ can be generated through a transformation $\tau \in \mathcal{T}$ for $G(\mathbf{A}, \mathbf{X})$, where $\mathcal{T}$ represents the whole augmentation function space. Concretely, the graph augmentation $\tau$ is jointly performed, considering both aspects of graph topology and feature, in alignment with prior research (Zhu et al., 2020). To realize topology-level augmentation, *edge removal* is employed, which randomly removes a certain ratio $p_e$ of edges within the original graph. In the aspect of feature, for feature matrix $\mathbf{X} \in \mathbb{R}^{N \times D}$, *node feature masking* randomly zeroizes a specific number $D \cdot p_f$ of feature channels, where $p_f$ is the resetting ratio.

**Basic Framework.**    As shown in Figure 2, our method follows the technical route of multi-view self-supervised learning, where the two various views $G'_A(\mathbf{A}'_A, \mathbf{X}'_A) = \tau_A(G)$ and $G'_B(\mathbf{A}'_B, \mathbf{X}'_B) = \tau_B(G)$ are generated based on two graph augmentation functions $\tau_A$ and $\tau_B$ randomly sampled from $\mathcal{T}$. The two augmented views are respectively fed into the shared network $f_\theta(\cdot)$, which is implemented via a graph neural network architecture, to acquire representations $\widetilde{\mathbf{H}}_A = [\tilde{\mathbf{h}}_1^A, ..., \tilde{\mathbf{h}}_N^A]^\top$ and $\widetilde{\mathbf{H}}_B = [\tilde{\mathbf{h}}_1^B, ..., \tilde{\mathbf{h}}_N^B]^\top$. For representation matrix $\widetilde{\mathbf{H}} = [\tilde{\mathbf{h}}_1, ..., \tilde{\mathbf{h}}_N]^\top$, each point $\tilde{\mathbf{h}}_i$ is normalized as $\mathbf{h}_i = \frac{\tilde{\mathbf{h}}_i - \mu(\widetilde{\mathbf{H}})}{\sigma(\widetilde{\mathbf{H}})}$, where $\mu(\widetilde{\mathbf{H}}) = \frac{1}{N} \sum_{i=1}^{N} \tilde{\mathbf{h}}_i \in \mathbb{R}^d$ denotes mean and $\sigma(\widetilde{\mathbf{H}}) = \sqrt{\frac{1}{N} \sum_{i=1}^{N} \left( \tilde{\mathbf{h}}_i - \mu(\widetilde{\mathbf{H}}) \right)^2} \in \mathbb{R}^d$ represents standard deviation. The centrally normalized matrices $\mathbf{H}_A = [\mathbf{h}_1^A, ..., \mathbf{h}_N^A]^\top$ and $\mathbf{H}_B = [\mathbf{h}_1^B, ..., \mathbf{h}_N^B]^\top$ can be obtained in this manner, respectively.

### 3.2 ANCHOR-NEIGHBORHOOD ALIGNMENT

One of the core components of multi-view self-supervised learning is to enhance cross-view consistency, thereby learning essential and invariant representations across multiple views. Technically, this consistency is achieved through the alignment of distinct augmented versions of the identical instance (*i.e.*, positive samples) in the representation space. Based on the notations in this paper, the alignment item can be formulated as

$$\mathcal{L}_{align} = \frac{1}{|\mathcal{V}|} \sum_{v \in \mathcal{V}} \|\mathbf{h}_v^A - \mathbf{h}_v^B\|_2^2. \tag{1}$$

However, in the context of graph self-supervised learning, naively treating two nodes from different augmented views as a positive pair is suboptimal. *On the one hand*, the graph harbors a wealth of structural information that characterizes the relationships between nodes, which can offer guidance for modeling the distribution of positive samples. Disregarding the potential role of the topological

structure in shaping self-supervised objectives and treating it solely as a regulator of message passing in GNNs lead to inadequate utilization of graph information. *On the other hand*, the positive sample pair is only acquired from various augmented versions of the same node, which poses a challenge in determining the strength of augmentation. The related studies have shown that increasing data augmentation can improve the quality of self-supervised representations, but overly strong augmentations introducing excessive disturbances result in a decline in performance (Tian et al., 2020). In other words, this *strict alignment* exhibits limited resilience to data augmentation.

Building upon the aforementioned considerations, we introduce a Topology-Guided Positive Sampling (TGPS) strategy to harness the potential of graph structure in self-supervised learning. Specifically, we establish a positive distribution centered around the anchor node based on graph structure and align positive samples derived from it with the anchor node:

$$\mathcal{L}_{\text{TGPS}} = \frac{1}{|\mathcal{V}|} \sum_{v \in \mathcal{V}} \mathbb{E}_{u \sim p(u|v)} \left[ \|\mathbf{h}_v^A - \mathbf{h}_u^B\|_2^2 + \|\mathbf{h}_v^B - \mathbf{h}_u^A\|_2^2 \right], \tag{2}$$

where the conditional probability $p(u|v)$ describes the distribution of positive samples of *anchor* node $v$. A good positive pair should contain two samples belonging to the same category, which means that the instances with the same label as the anchor node $v$ should be sampled from the conditional distribution $p(u|v)$. Graphs such as social networks are typically constructed under the homophily assumption, which potentially indicates that the nodes within small topological distances tend to share the same labels with a high probability. Therefore, using structural context of the anchor node to model the conditional distribution is a decent option.

Specifically, graph diffusion is employed to characterize the structural context distribution, which can well describe the affinity between two nodes. The affinity matrix $\mathbf{S}$ is formulated as follows:

$$\mathbf{S} = \mathbf{r} \odot \hat{\mathbf{S}}, \quad \hat{\mathbf{S}} = \sum_{k=0}^{K} e^{-tk} (\hat{\mathbf{D}}^{-1} \hat{\mathbf{A}})^k, \tag{3}$$

where $\hat{\mathbf{D}}$ is the diagonal degree matrix of $\hat{\mathbf{A}} = \mathbf{A} + \mathbf{I}_N$, $K$ decides the hop number of utilized neighbors, $t$ is a damping coefficient, $\odot$ denotes the broadcasted hadamard product, and $\mathbf{r} \in \mathbb{R}^{N \times 1}$ in which $r_{i1} = 1/(\sum_{j=1}^{N} \hat{S}_{ij})$. The affinity matrix $\mathbf{S}$ based on graph diffusion describes the weighted sum of landing probabilities of multiple random walking matrices under various steps . $e^{-tk}$ denotes the weighted coefficient, where the smaller the value of $t$, the higher the affinity between two nodes with large topological distance. Each row of $\mathbf{S}$, presenting a context distribution, actually instantiates a conditional probability distribution $p(u|v)$. From the perspective of probabilistic modeling, the value of $K$ determines maximum sampling range, while $t$ can influence probability mass. For node $v_i$, the probability distribution of its positive samples is

$$p(v_j|v_i) = S_{ij}. \tag{4}$$

The conditional probability $p(u|v)$ in Eq. (2) can be instantiated by $\mathbf{S}$, which eventually realize weighted Anchor-Neighborhood Alignment (ANA) in representation space:

$$\mathcal{L}_{ana} = \frac{1}{N} \sum_{i=1}^{N} \sum_{v_j \in \mathcal{N}_i^K} S_{ij} \cdot \left[ \|\mathbf{h}_i^A - \mathbf{h}_j^B\|_2^2 + \|\mathbf{h}_i^B - \mathbf{h}_j^A\|_2^2 \right], \tag{5}$$

where $\mathcal{N}_i^K$ indicates the set of neighbors of anchor node $v_i$ within $K$ hops, including node $v_i$ itself.

Nevertheless, direct calculation of Eq. (5) will result in space complexity of $\mathcal{O}(N^2 d)$. To this end, we turn to optimize an upper bound of Eq. (5):

$$\mathcal{L}_{\text{ANA}} = -\frac{2}{N} \cdot tr(\mathbf{H}_A^\top \mathbf{S} \mathbf{H}_B + \mathbf{H}_B^\top \mathbf{S} \mathbf{H}_A), \tag{6}$$

where $tr(\cdot)$ denotes the matrix trace. Refer to Appendix B for the derivation from Eq. (5) to Eq. (6).

The matrix $\mathbf{S}$ can be calculated in advance of the formal training, which does not increase additional computational burden during training. Besides, the positive sampling process is converted to straightforward matrix multiplication operation, which avoids explicit a sampling process and does not need extra storage for positive samples. These advantages render our approach highly efficient in terms of computation and storage.

### 3.3 ISOTROPIC CONSTRAINTS

Our research focuses on fully unleashing the potential of graph self-supervised learning, which inherently necessitates enhancing expressive capacity of the models and diversity of node representations. Under-expressed representations fail to adequately span the entire representation space, giving rise to the notorious issue of dimension collapse in the realm of self-supervised learning. In existing literature, a prevailing viewpoint on dimensional collapse is that distinct dimensions manifest significant correlations, which inevitably convey coupled and redundant information. Beyond this prevalent opinion, as depicted in Figure 1(c), another potential manifestation of dimensional collapse is that the representations exhibit certain distributional disparities along various directions. Along the direction of a *unit projection vector* $\mathbf{q}_j^A \in \mathbb{R}^d$, the projections of the $N$ points in the representation matrix $\mathbf{H}_A \in \mathbb{R}^{N \times d}$ can be described as

$$\mathbf{z}_j^A = \mathbf{H}_A \mathbf{q}_j^A \in \mathbb{R}^N. \tag{7}$$

When the projections of representations along any arbitrary directions display consistent distribution, the dimensional collapse issue naturally dissipates, that is, node representations demonstrate isotropic characteristics. Different perspectives determine distinct approaches. Therefore, a corresponding strategy to mitigate dimension collapse is to impose Isotropic Constraints (IC) on the node representations, compelling them to exhibit consistent distribution along distinct directions and facilitating diverse representations. One ensuing challenge is how to characterize the distribution of representations along a specific direction.

**Definition 1** (Central Moment). *The $m$-th central moment of a one-dimensional random variable $X$ can be defined as*

$$CM_m(X) = \mathbb{E}[(X - \mathbb{E}[X])^m], \tag{8}$$

*where $\mathbb{E}[\cdot]$ is the expectation operator.*

Different orders of moments can characterize the spread and shapes of a random variable from distinct aspects (Grimmett & Stirzaker, 2020). Thus, an ideal strategy for isotropic constraints is to make various moments of projections of representations along all spatial directions consistent respectively. However, simultaneously traversing all directions within the entire space is infeasible. To this end, we employ a strategy of randomly sampling $n$ unit vectors at each optimization step, thereby reducing the distributional disparities of representations along these $n$ directions. The $N$ projected points along each direction can be regarded as $N$ empirical observations of a one-dimensional variable. As a result, the following objective for isotropic constraints is constructed:

$$\mathcal{L}_{ic}^* = \sum_{m=2}^{M} \alpha_m \cdot var\left(\{CM_m(\mathbf{H}_* \mathbf{q}_j^*)|j=1,\ldots,n\}\right), \tag{9}$$

where $* \in \{A, B\}$[1], $\{\mathbf{q}_j^*|j=1,\ldots,n\}$ are $n$ random unit projection vectors, $\{\mathbf{H}_* \mathbf{q}_j^*|j=1,\ldots,n\}$ represent the corresponding collection of $n$ groups of projected points, $var(\cdot)$ indicates the variance, $\alpha_m$ denotes a weighted factor, and $M$ is maximum order of central moments. In Eq. (9), the inner $CM_m(\cdot)$ characterizes the distribution of node representations along a specific direction; the outer $var(\cdot)$ measures the distribution difference between projections along distinct directions, minishing which contributes to achieving isotropic distribution. However, considering multiple moments will result in a substantial computational burden. To further mitigate complexity, only the 2-nd central moment (*i.e.*, variance) is considered in practice, which transforms Eq. (9) into $\mathcal{L}_{ic}^* = var(\{var(\mathbf{H}_* \mathbf{q}_j^*)|j=1,\ldots,n\})$. Taking views $A$ and $B$ into consideration, the overall objective function based on isotropic constraints can be formulated as

$$\mathcal{L}_{\mathtt{IC}} = \mathcal{L}_{ic}^A + \mathcal{L}_{ic}^B. \tag{10}$$

To expedite the convergence speed of the training process, an intuitive strategy is to maximize the angular separation between the $n$ projection vectors. A feasible approach is to ensure pairwise orthogonality among the $n$ vectors. To this end, the number $n$ of projection vectors is set to $d$. For each view, we can construct an orthogonal matrix $\mathbf{Q}_* = [\mathbf{q}_1^*, \ldots, \mathbf{q}_d^*] \in \mathbb{R}^{d \times d}$ to generate $d$ unit projection vectors, which actually performs a rotation transformation on the original representations $\mathbf{H}_*$. In practice, the $d$ unit projection vectors can be obtained through the QR factorization (Gu & Eisenstat, 1996) of a randomly generated matrix with a size of $d \times d$. Besides, two groups of projection vectors for views $A$ and $B$ are independently generated to traverse as many spatial directions as possible at each training step.

---

[1] In the remainder of this paper, $*$ denotes either $A$ or $B$.

**Relation with decorrelation-based methods.** The decorrelation-based methods such as CCA-SSG (Zhang et al., 2021) mitigate dimensional collapse issue by decoupling various representation dimensions through a decorrelation loss $\mathcal{L}_{\text{DEC}} = \|\frac{1}{N}\mathbf{H}_A^\top\mathbf{H}_A - \mathbf{I}\|_F^2 + \|\frac{1}{N}\mathbf{H}_B^\top\mathbf{H}_B - \mathbf{I}\|_F^2$. This class of methods can be viewed as a specific case of our approach under particular settings.

**Proposition 1.** *If only the 2-nd central moment is considered and the $d$ projection vectors are fixed to $d$ eigenvectors of the covariance matrix $\mathbf{\Sigma}_* = \frac{1}{N}\mathbf{H}_*^\top\mathbf{H}_*$ at each training step, optimizing the decorrelation loss will be equivalent to imposing isotropic constraints, that is, minimizing $\|\frac{1}{N}\mathbf{H}_*^\top\mathbf{H}_* - \mathbf{I}\|_F^2$ is equivalent to minishing $var(\{var(\mathbf{H}_*\mathbf{q}_j^*)|j = 1, \dots, d\})$.*

*Proof.* Refer to Appendix C. $\square$

**Relation with information entropy maximization.** In deep learning, information entropy provides a description about information amount and redundancy within the representations.

**Proposition 2.** *When representations conform to the same distribution along any spatial direction (that is, achieving isotropy), its Gaussian entropy will be maximized.*

*Proof.* Refer to Appendix D. $\square$

**Relation with negative-sampling-based methods.** An influential study (Wang & Isola, 2020) decomposes the classical *InfoNCE* loss into alignment and uniformity terms and demonstrates that, under the mutual exclusion effect of negative samples, representations tend to exhibit a uniform distribution on the unit hypersphere. Obviously, under this condition, the representations will display consistent distribution along any projection direction, that is, achieving isotropy.

### 3.4 OVERALL OBJECTIVE FUNCTION

ANA serves as a cornerstone module, endowing the neural model with the ability to learn augmentation-invariant representations and capture structural information. IC can reinforce the expressive power of the model and improve the diversity of representations. The two terms complement each other, forming a comprehensive self-supervised objective:

$$\mathcal{L} = \mathcal{L}_{\text{ANA}} + \lambda \cdot \mathcal{L}_{\text{IC}}, \tag{11}$$

where $\lambda$ denotes a balancing factor. The overall algorithm flow is presented in Algorithm 1.

## 4 EXPERIMENTS

In this section, we carry out extensive experiments to evaluate the effectiveness and efficiency of our method and provide detailed ablation studies and visual analysis to gain a deeper understanding of its underlying principles. The code is available at an anonymous repository: https://github.com/AnonymousSubConf/ANAIC.

### 4.1 DATASETS AND EXPERIMENTAL SETUP

**Datasets.** To evaluate the proposed approach, six widely recognized benchmark datasets are employed for empirical studies, covering three citation networks Cora, Citeseer, and Pubmed (Sen et al., 2008), two co-purchase networks Amazon-Computers and Amazon-Photo (Shchur et al., 2019), and one co-authorship network Coauthor-CS (Shchur et al., 2019).

**Experimental Setup.** The model is implemented by Graph Convolutional Network (GCN) (Kipf & Welling, 2016a), whose parameters are initialized via Xavier initialization (Glorot & Bengio, 2010) and trained with Adam optimizer (Kingma & Ba, 2017). All experiments are performed on a TITAN RTX GPU with 24 GB of memory. The representations are initially learned by our approach in an unsupervised manner and subsequently evaluated using a simple linear classifier.

### 4.2 COMPARISON WITH STATE-OF-THE-ART BASELINES

In this subsection, we conduct a comparative analysis, pitting our method against state-of-the-art baselines with regard to both effectiveness and efficiency.

Table 1: Node classification accuracy with standard deviation in percentage across six experimental datasets. The "**Input**" column represents the data utilized during the training phase, and **Y** denotes labels. "OOM" means Out-Of-Memory.

| | Algorithm | Input | Cora | Citeseer | Pubmed | Computers | Photo | Coauthor-CS |
|---|---|---|---|---|---|---|---|---|
| | MLP | X, Y | $57.8 \pm 0.2$ | $54.2 \pm 0.1$ | $72.8 \pm 0.2$ | $79.81 \pm 0.06$ | $86.36 \pm 0.08$ | $91.32 \pm 0.11$ |
| | GCN | X, A, Y | 81.5 | 70.3 | 79.0 | $86.51 \pm 0.54$ | $92.42 \pm 0.22$ | $93.03 \pm 0.31$ |
| | GAT | X, A, Y | $83.0 \pm 0.7$ | $72.5 \pm 0.7$ | $79.0 \pm 0.3$ | $86.93 \pm 0.29$ | $92.56 \pm 0.35$ | $92.31 \pm 0.24$ |
| Unsupervised | DeepWalk | A | $68.5 \pm 0.5$ | $49.8 \pm 0.2$ | $66.2 \pm 0.7$ | $85.68 \pm 0.06$ | $89.44 \pm 0.11$ | $84.61 \pm 0.22$ |
| | GAE | X, A | $72.1 \pm 0.5$ | $66.5 \pm 0.4$ | $71.8 \pm 0.6$ | $85.27 \pm 0.19$ | $91.62 \pm 0.13$ | $90.01 \pm 0.71$ |
| | GMI | X, A | $83.0 \pm 0.3$ | $72.4 \pm 0.1$ | $79.9 \pm 0.2$ | $82.21 \pm 0.31$ | $90.68 \pm 0.17$ | OOM |
| | GRACE | X, A | $81.9 \pm 0.4$ | $71.3 \pm 0.3$ | $80.1 \pm 0.2$ | $86.53 \pm 0.28$ | $92.24 \pm 0.17$ | $92.98 \pm 0.05$ |
| | GCA | X, A | $81.7 \pm 0.3$ | $71.1 \pm 0.4$ | $79.5 \pm 0.5$ | $87.85 \pm 0.31$ | $92.49 \pm 0.09$ | $93.10 \pm 0.01$ |
| | GraphMAE | X, A | $84.2 \pm 0.4$ | $73.4 \pm 0.4$ | $81.1 \pm 0.4$ | $88.12 \pm 0.30$ | $92.97 \pm 0.21$ | $93.03 \pm 0.16$ |
| | CCA-SSG | X, A | $84.2 \pm 0.4$ | $73.1 \pm 0.3$ | $81.6 \pm 0.4$ | $88.74 \pm 0.28$ | $93.14 \pm 0.14$ | $93.31 \pm 0.22$ |
| | G-BT | X, A | $84.0 \pm 0.4$ | $73.0 \pm 0.3$ | $80.7 \pm 0.4$ | $88.14 \pm 0.33$ | $92.63 \pm 0.44$ | $92.95 \pm 0.17$ |
| | InfoGCL | X, A | $83.5 \pm 0.3$ | $73.5 \pm 0.4$ | $79.1 \pm 0.2$ | - | - | - |
| | SUGRL | X, A | $83.4 \pm 0.5$ | $73.0 \pm 0.4$ | $81.9 \pm 0.3$ | $88.12 \pm 0.21$ | $92.94 \pm 0.13$ | $92.87 \pm 0.14$ |
| | MVGRL | X, A | $83.7 \pm 0.6$ | $\mathbf{73.6 \pm 0.3}$ | $79.9 \pm 0.2$ | $87.52 \pm 0.11$ | $91.74 \pm 0.07$ | $92.11 \pm 0.12$ |
| | GGD | X, A | $83.5 \pm 0.4$ | $73.1 \pm 0.5$ | $80.7 \pm 0.7$ | - | - | - |
| | DGI | X, A | $82.3 \pm 0.6$ | $71.8 \pm 0.7$ | $76.8 \pm 0.6$ | $83.95 \pm 0.47$ | $91.61 \pm 0.22$ | $92.15 \pm 0.63$ |
| | ANA-IC (ours) | X, A | $\mathbf{84.5 \pm 0.6}$ | $\mathbf{73.6 \pm 0.6}$ | $\mathbf{82.9 \pm 0.4}$ | $\mathbf{88.82 \pm 0.34}$ | $\mathbf{93.20 \pm 0.17}$ | $\mathbf{94.54 \pm 0.21}$ |

**Performance Evaluation.** Here, we assess the effectiveness of our approach by conducting a comparative analysis against state-of-the-art baselines in the context of node classification task under the simple linear evaluation. The average classification accuracy with standard deviation over 20 random initialization is reported for each dataset. We compare our approach with unsupervised methods covering DeepWalk (Perozzi et al., 2014), GAE (Kipf & Welling, 2016b), DGI (Veličković et al., 2018), GMI (Peng et al., 2020), GRACE (Zhu et al., 2020), GCA (Zhu et al., 2021), G-BT (Bielak et al., 2022), InfoGCL (Xu et al., 2021), SUGRL (Mo et al., 2022), GraphMAE (Hou et al., 2022), CCA-SSG (Zhang et al., 2021), GGD (ZHENG et al., 2022) and MVGRL (Hassani & Khasahmadi, 2020). Additionally, some supervised models, including multi-layer perceptron (MLP), GCN (Kipf & Welling, 2016a), and GAT (Veličković et al., 2017), are also added as baselines. Following the predecessors (Zhu et al., 2021; Zhang et al., 2021), we employ the publicly available splits on Cora, Citeseer, and Pubmed and a 1:1:8 split for training/validation/testing on the remaining three datasets. To ensure a fair comparison, in cases where other methods do not use the same dataset splits as ours, we obtain pertinent results by referring to their officially released source code. Table 1 presents the classification results for the six datasets. It is evident that our method demonstrates outstanding performance across all six datasets, consistently outperforming both the unsupervised and fully supervised baselines by substantial margins. These high-performance results underscore the effectiveness and superiority of our approach.

**Efficiency Comparison.** Please refer to Appendix E.1.

### 4.3 ABLATION STUDY AND SENSITIVITY ANALYSIS

In this subsection, we conduct ablation studies on key components and sensitivity analysis for crucial hyperparameters to gain deeper insights into our approach.

**Effect of Anchor-Neighborhood Alignment, Isotropic Constraints, Data Augmentation, and Orthogonality Between Projection Vectors.** As summarized in Table 2, we sequentially eliminate the critical components of our approach to analyze their influence on node classification accuracy. The exclusion of the IC term results in a noticeable deterioration in performance, which potentially gives rise to dimensional collapse issue and constrains the richness of node representations. As expected, optimizing solely the IC term causes the model to learn representations that are diverse but ultimately

Table 2: Ablation studies on key components of our method. "w/o" indicates "without".

| Variants | Cora | Pubmed | CS |
|---|---|---|---|
| ANA-IC (baseline) | 84.5 | 82.9 | 94.54 |
| w/o $\mathcal{L}_{\text{IC}}$ | 79.3 | 74.1 | 92.74 |
| w/o $\mathcal{L}_{\text{ANA}}$ | 56.4 | 55.8 | 27.22 |
| w/o Augmentation | 78.6 | 81.8 | 93.36 |
| w/o orthogonality | 84.1 | 82.4 | 94.43 |

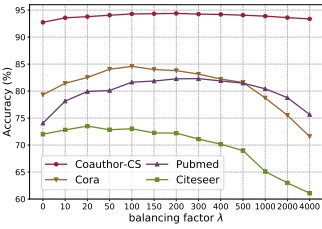

Figure 3: The node classification performance under varying balancing factors.

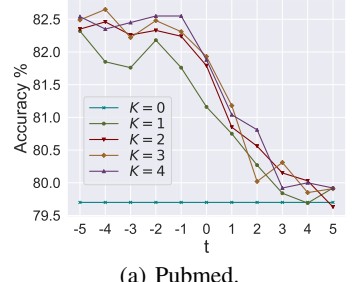

(a) Pubmed.

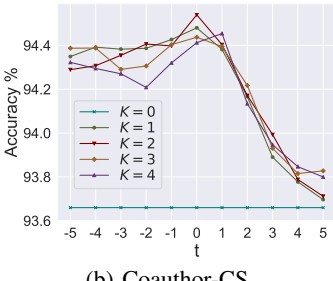

(b) Coauthor-CS.

Figure 4: The effects under various combinations of $K$ and $t$.

devoid of meaningful content, resulting in poor performance. For ANA, the self-supervised model can maintain its performance without requiring data augmentation, as it effectively acquires valuable information from structural context. When imposing isotropic constraints on representations, the performance remains highly significant even without orthogonalizing the projection vectors. This result is attributed to the specific angular differences inherent among randomly generated projection vectors and can greatly streamline our approach.

**Influence of Balancing Factor.** We investigate the variation of classification accuracy with respect to the balancing coefficient $\lambda$ in Eq. (11), as illustrated in Figure 3. We can observe that the performance exhibits an initial upward trend followed by a subsequent decrease as $\lambda$ increases. When $\lambda$ is minor, the IC term cannot effectively serve its function in enhancing diversity of representations. Conversely, when $\lambda$ is excessively large, there is an overemphasis on isotropic constraints, resulting in abundant yet meaningless representations. The performance of our method benefits from an appropriate value of $\lambda$. Fortunately, the peak performance is achieved when $\lambda$ falls within the range of 50 to 200 across most experimental datasets, which significantly simplifies the process of hyperparameter tuning when applying our method to a new dataset.

**Impact of Neighborhood Size and Weight.** We explore the influence of $K$ and $t$ in Eq. (3), which determines the maximum sampling range and influences probability mass, respectively. Figure 4 shows the performance under various values of $K$ and $t$ on Coauthor-CS and Pubmed. When $K$ is equal to 0, the anchor-neighborhood alignment in Eq. (5) degenerates into strict alignment in Eq. (1), which obtains the worst results on both datasets. For a particular value of $K > 0$, the performance benefits from a proper value of $t$. When $t$ is too small (*e.g.*, $-5$ or $-4$), it overemphasizes the alignment between anchor node and neighborhood nodes, leading to suboptimal performance. When $t$ is too large (that is, $e^{-tk} \rightarrow 0$ for $k > 0$), it approximately degenerates into strict alignment in Eq. (1). Besides, some hyperparameter combinations, such as $K = 2$ and $t = 0$, consistently perform well in all datasets, facilitating hyperparameter tuning in real-world scenarios or new datasets. Overall, the experimental results demonstrate that constructing a positive sampling distribution based on structural context helps improve performance.

The additional experiments are placed in Appendix E.

## 5 CONCLUSION

In this paper, we have introduced a comprehensive self-supervised learning framework comprised of two complementary components. The objective is to train expressive neural models capable of effectively harnessing a vast reservoir of unlabeled graph data. Firstly, a positive sampling strategy based on structural context is designed to realize weighted alignment between anchor node and neighborhood nodes, which enhances adequate exploration for graph information and endows the model with stronger structure-ware ability. Secondly, we revisit the appearance of under-expressed representations and, correspondingly, propose a novel strategy of isotropic constraints to improve diversity of representations and overcome tricky dimensional collapse. We believe that our research can propel the study of dimension collapse forward, which is a crucial topic in multi-view self-supervised learning. Due to no reliance on mutual information estimator, additional projection heads, and negative samples, our approach exhibits remarkable efficiency in terms of both training time and resource consumption. Thorough experiments substantiate the effectiveness and efficiency of our approach.

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

Table 3: Statistics of the experimental datasets.

| Dataset | Nodes | Edges | Features | Classes |
|---|---|---|---|---|
| Cora | 2,708 | 5,429 | 1,433 | 7 |
| Citeseer | 3,327 | 4,732 | 3,703 | 6 |
| Pubmed | 19,717 | 44,338 | 500 | 3 |
| Amazon-Computers | 13,752 | 245,861 | 767 | 10 |
| Amazon-Photo | 7,650 | 119,081 | 745 | 8 |
| Coauthor-CS | 18,333 | 81,894 | 6,805 | 15 |

## A  STATISTICS OF THE EXPERIMENTAL DATASETS

The statistics of the experimental datasets are summarized in Table 3. The details of the datasets are as follows:

- **Cora, Citeseer, and Pubmed** are citation networks where nodes represent documents and edges signify citation connections. Each document is assigned a class label denoting its subject category and is linked to a bag-of-words feature vector.

- **Amazon-Computers and Amazon-Photo** are two graphs derived from Amazon, capturing co-purchase relationships. Nodes represent products, and edges connect nodes that are frequently purchased in tandem. Each node is characterized by a sparse bag-of-words feature extracted from product reviews, and its label designates the product category.

- **Coauthor-CS** is an academic network wherein nodes represent authors, and edges signify co-authorship relationship. Authors are connected if they have collaborated on a research paper together.

## B  DERIVATION FROM EQ. (5) TO EQ. (6) IN THE MAIN TEXT

In this section, we provide the detailed derivation from Eq. (5) to Eq. (6) in the main text. We first introduce two lemmas required for derivation and then formally give the derivation.

### B.1  TWO IMPORTANT LEMMAS

**Lemma 1.** *For a random variable $x$, with variance $\mathbb{D}[x]$ and mean $\mathbb{E}[x]$, it is satisfied that*

$$\mathbb{D}[x] = \mathbb{E}[x^2] - (\mathbb{E}[x])^2. \tag{12}$$

*Proof.* Let $u = \mathbb{E}[x]$, thus $\mathbb{D}[x] = \mathbb{E}[(x-u)^2] = \mathbb{E}[x^2 - 2ux + u^2] = \mathbb{E}[x^2] - 2u\mathbb{E}[x] + u^2 = \mathbb{E}[x^2] - (\mathbb{E}[x])^2.$ □

**Lemma 2.** *For matrix $\mathbf{A} \in \mathbb{R}^{N \times K}$ and matrix $\mathbf{B} \in \mathbb{R}^{N \times K}$,*

$$tr(\mathbf{A}\mathbf{B}^\top) = tr(\mathbf{A}^\top \mathbf{B}). \tag{13}$$

*Proof.*

$$
\begin{aligned}
&tr(\mathbf{A}\mathbf{B}^\top) \\
&= \sum_{i=1}^{N} \sum_{j=1}^{K} A_{ij}(B^\top)_{ji} \\
&= \sum_{i=1}^{N} \sum_{j=1}^{K} A_{ij}B_{ij} \\
&= \sum_{j=1}^{K} \sum_{i=1}^{N} A_{ij}B_{ij} \\
&= \sum_{j=1}^{K} \sum_{i=1}^{N} (A^\top)_{ji}B_{ij} \\
&= tr(\mathbf{A}^\top\mathbf{B}).
\end{aligned}
\tag{14}
$$

$\square$

### B.2 DERIVATION FROM EQ. (5) TO EQ. (6) IN THE MAIN TEXT

We split

$$
\frac{1}{N} \sum_{i=1}^{N} \sum_{v_j \in \mathcal{N}_i^K} S_{ij} \cdot \left[ \|\mathbf{h}_i^A - \mathbf{h}_j^B\|_2^2 + \|\mathbf{h}_i^B - \mathbf{h}_j^A\|_2^2 \right]
$$

into two terms:

$\frac{1}{N} \sum_{i=1}^{N} \sum_{v_j \in \mathcal{N}_i^K} S_{ij} \cdot \|\mathbf{h}_i^A - \mathbf{h}_j^B\|_2^2$ and $\frac{1}{N} \sum_{i=1}^{N} \sum_{v_j \in \mathcal{N}_i^K} S_{ij} \cdot \|\mathbf{h}_i^B - \mathbf{h}_j^A\|_2^2$.

According to Lemma 1, we can know that $\frac{1}{N} \sum_{i=1}^{N} (H_{ik}^A)^2 = 1$ and $\frac{1}{N} \sum_{i=1}^{N} (H_{ik}^B)^2 = 1$. Assuming the maximal element in matrix $\mathbf{S}$ is $S_{max}$, the derivation of first term is as follows:

$$
\begin{aligned}
&\frac{1}{N} \sum_{i=1}^{N} \sum_{v_j \in \mathcal{N}_i^K} S_{ij} \cdot \|\mathbf{h}_i^A - \mathbf{h}_j^B\|_2^2 \\
=&\frac{1}{N} \sum_{i=1}^{N} \sum_{j=1}^{N} S_{ij} \cdot \left\|\mathbf{h}_i^A - \mathbf{h}_j^B\right\|_2^2 \\
=&\frac{1}{N} \sum_{i=1}^{N} \sum_{j=1}^{N} (\mathbf{h}_i^{A\top}\mathbf{h}_i^A - 2 \cdot \mathbf{h}_i^{A\top}\mathbf{h}_j^B + \mathbf{h}_j^{B\top}\mathbf{h}_j^B) \cdot S_{ij} \\
=&\frac{1}{N} \sum_{i=1}^{N} (\sum_{j=1}^{N} S_{ij})\mathbf{h}_i^{A\top}\mathbf{h}_i^A + \frac{1}{N} \sum_{j=1}^{n} (\sum_{i=1}^{N} S_{ij})\mathbf{h}_j^{B\top}\mathbf{h}_j^B - \frac{2}{N} \sum_{i=1}^{N} \sum_{j=1}^{N} \mathbf{h}_i^{A\top}\mathbf{h}_j^B S_{ij} \\
\leq&\frac{1}{N} \sum_{i=1}^{N} (\sum_{j=1}^{N} S_{max})\mathbf{h}_i^{A\top}\mathbf{h}_i^A + \frac{1}{N} \sum_{j=1}^{n} (\sum_{i=1}^{N} S_{max})\mathbf{h}_j^{B\top}\mathbf{h}_j^B - \frac{2}{N} \sum_{i=1}^{N} \sum_{j=1}^{N} \mathbf{h}_i^{A\top}\mathbf{h}_j^B S_{ij} \\
=&S_{max} \sum_{i=1}^{N} \mathbf{h}_i^{A\top}\mathbf{h}_i^A + S_{max} \sum_{j=1}^{N} \mathbf{h}_j^{B\top}\mathbf{h}_j^B - \frac{2}{N} \sum_{i=1}^{N} \sum_{j=1}^{N} \mathbf{h}_i^{A\top}\mathbf{h}_j^B S_{ij} \\
=&S_{max} \sum_{k=1}^{d} \sum_{i=1}^{N} (H_{ik}^A)^2 + S_{max} \sum_{k=1}^{d} \sum_{j=1}^{N} (H_{jk}^B)^2 - \frac{2}{N} \sum_{i=1}^{N} \sum_{j=1}^{N} \mathbf{h}_i^{A\top}\mathbf{h}_j^B S_{ij}
\end{aligned}
$$

$$=NS_{max}\sum_{k=1}^{d}\frac{1}{N}\sum_{i=1}^{N}(H_{ik}^A)^2 + NS_{max}\sum_{k=1}^{d}\frac{1}{N}\sum_{j=1}^{N}(H_{jk}^B)^2 - \frac{2}{N}\sum_{i=1}^{N}\sum_{j=1}^{N}\mathbf{h}_i^{A\top}\mathbf{h}_j^B S_{ij}$$

$$=2dNS_{max} - \frac{2}{N}\sum_{i=1}^{N}\sum_{j=1}^{N}\mathbf{h}_i^{A\top}\mathbf{h}_j^B S_{ij}$$

$$=2dNS_{max} - \frac{2}{N}\sum_{i=1}^{N}\mathbf{h}_i^{A\top}(\sum_{j=1}^{N}\mathbf{h}_j^B S_{ij})$$

$$=2dNS_{max} - \frac{2}{N}\sum_{i=1}^{N}\mathbf{h}_i^{A\top}(\mathbf{SH}^B)_i$$

$$=2dNS_{max} - \frac{2}{N}\cdot tr(\mathbf{H}_A(\mathbf{SH}_B)^\top)$$

$$=2dNS_{max} - \frac{2}{N}\cdot tr(\mathbf{H}_A^\top\mathbf{SH}_B)$$

Ignoring the constant term, we accomplish the derivation of the first term. Symmetrically, we can easily obtain the derivation of another term. Hence, we complete the overall derivation.

## C    PROOF OF PROPOSITION 1

To prove the Proposition 1, we require the following Property 1 as a fundamental prerequisite.

**Property 1.** *For a covariance matrix* $\mathbf{\Sigma} = \frac{1}{N}\mathbf{H}^\top\mathbf{H} \in \mathbb{R}^{d\times d}$ *derived from a centrally normalized matrix* $\mathbf{H} = [\mathbf{h}_1,\ldots,\mathbf{h}_N]^\top \in \mathbb{R}^{N\times d}$, *which has* $d$ *eigenvalues* $[\lambda_1,\lambda_2,\ldots,\lambda_d]$ *corresponding to* $d$ *eigenvectors* $[\mathbf{q}_1,\mathbf{q}_2,\ldots,\mathbf{q}_d]$, *the variance (i.e., 2-nd central moment) of the data* $\mathbf{H}$ *along the* $k$-th *principal direction (i.e., the direction defined by* $\mathbf{q}_k$) *amounts to* $\lambda_k$.

*Proof.* The covariance matrix of a representation matrix $\mathbf{H} = [\mathbf{h}_1,\ldots,\mathbf{h}_N]^\top \in \mathbb{R}^{N\times d}$, having been normalized to 0-mean and 1-variance along sample direction, is $\mathbf{\Sigma} = \frac{1}{N}\mathbf{H}^\top\mathbf{H}$. After eigendecomposing $\mathbf{\Sigma}$, we can acquire $d$ unit orthogonal eigenvectors $[\mathbf{q}_1,\ldots,\mathbf{q}_d]$ corresponding to eigenvalues $[\lambda_1,\ldots,\lambda_d]$, respectively. Due to $\frac{1}{N}\mathbf{H}^\top\mathbf{H}\mathbf{q}_k = \lambda_k\mathbf{q}_k$, it can be known that

$$\frac{1}{N}\mathbf{q}_k^\top\mathbf{H}^\top\mathbf{H}\mathbf{q}_k = \lambda_k\mathbf{q}_k^\top\mathbf{q}_k = \lambda_k. \tag{15}$$

Taking a principal direction $\mathbf{q}_k$ as explanation, the projection point of a sample $\mathbf{h}_i$ onto this direction is $z_i = \mathbf{q}_k^\top\mathbf{h}_i$, and the mean of all projections is

$$\bar{z} = \frac{1}{N}\sum_{i=1}^{N}z_i = \frac{1}{N}\sum_{i=1}^{N}\mathbf{q}_k^\top\mathbf{h}_i = \frac{1}{N}\mathbf{q}_k^\top(\sum_{i=1}^{N}\mathbf{h}_i) = 0. \tag{16}$$

Furthermore, along the principal direction $\mathbf{q}_k$, the variance is

$$\begin{aligned}
&\frac{1}{N}\sum_{i=1}^{N}(z_i - \bar{z})^2\\
&=\frac{1}{N}\sum_{i=1}^{N}\mathbf{q}_k^\top\mathbf{h}_i\mathbf{h}_i^\top\mathbf{q}_k\\
&=\frac{1}{N}\mathbf{q}_k^\top(\sum_{i=1}^{N}\mathbf{h}_i\mathbf{h}_i^\top)\mathbf{q}_k\\
&=\frac{1}{N}\mathbf{q}_k^\top\mathbf{H}^\top\mathbf{H}\mathbf{q}_k\\
&=\lambda_k.
\end{aligned} \tag{17}$$

The above equation illustrates that the variance of data $\mathbf{H}$ along the direction $\mathbf{q}_k$ amounts to $\lambda_k$. The proof is concluded.                                                                    $\square$

For convenience, we restate the Proposition 1:

**Proposition 1.** *If only the 2-nd central moment is considered and the $d$ projection vectors are fixed to $d$ eigenvectors of the covariance matrix $\Sigma_* = \frac{1}{N}\mathbf{H}_*^\top\mathbf{H}_*$ at each training step, optimizing the decorrelation loss will be equivalent to imposing isotropic constraints, that is, minimizing $\|\frac{1}{N}\mathbf{H}_*^\top\mathbf{H}_* - \mathbf{I}\|_F^2$ is equivalent to minishing $var(\{var(\mathbf{H}_*\mathbf{q}_j^*)|j=1,\ldots,d\})$.*

*Proof.* For the covariance matrix $\Sigma_*$ with $d$ eigenvalues $[\lambda_1, \lambda_2, \ldots, \lambda_d]$ corresponding to $d$ eigenvectors $[\mathbf{q}_1, \mathbf{q}_2, \ldots, \mathbf{q}_d]$, $tr(\Sigma_*) = \sum_{j=1}^d 1 = d = \sum_{j=1}^d \lambda_j$, where $tr(\cdot)$ denotes the trace of a matrix. Besides, $tr(\Sigma_*^2) = \sum_{j=1}^d \lambda_j^2$. According to Property 1, $var(\mathbf{H}_*\mathbf{q}_j^*) = \lambda_j$. Given that $\bar{\lambda} = \frac{1}{d}\sum_{j=1}^d \lambda_j = 1$, we can know that $\qquad\qquad\square$

$$
\begin{aligned}
&var(\{var(\mathbf{H}_*\mathbf{q}_j^*)|j=1,\ldots,d\})\\
=&var(\{\lambda_j|j=1,\ldots,d\})\\
=&\frac{1}{d}\sum_{j=1}^d(\bar{\lambda}-\lambda_j)^2\\
=&\frac{1}{d}\cdot d\cdot\bar{\lambda}^2 - \frac{2}{d}\cdot\bar{\lambda}\cdot\sum_{j=1}^d\lambda_j + \frac{1}{d}\sum_{j=1}^d\lambda_j^2\\
=&\bar{\lambda}^2 - 2\cdot\bar{\lambda} + \frac{1}{d}\sum_{j=1}^d\lambda_j^2\\
=&\frac{1}{d}\sum_{j=1}^d\lambda_j^2 - 1\\
=&\frac{1}{d}tr(\Sigma_*^2) - 1.
\end{aligned}
\tag{18}
$$

As $\Sigma_* - \mathbf{I}$ is a symmetric matrix, we can know that

$$
\begin{aligned}
&\|\frac{1}{N}\mathbf{H}_*^\top\mathbf{H}_* - \mathbf{I}\|_F^2\\
=&\|\Sigma_* - \mathbf{I}\|_F^2\\
=&tr((\Sigma_* - \mathbf{I})^\top(\Sigma_* - \mathbf{I}))\\
=&tr(\Sigma_*^2) - 2\cdot tr(\Sigma_*) + tr(\mathbf{I})\\
=&tr(\Sigma_*^2) - d.
\end{aligned}
\tag{19}
$$

Compare the results of Eq. (18) and Eq. (19), we can observe that they differ by a constant of $d$. Therefore, minimizing one is equivalent to the other. We conclude the proof.

## D  PROOF OF PROPOSITION 2

**Proposition 2.** *When representations conform to the same distribution along any spatial direction (that is, achieving isotropy), its Gaussian entropy will be maximized.*

*Proof.* Under the Gaussian assumption, the information entropy (Ahmed & Gokhale, 1989) of representations with the covariance matrix $\Sigma_*$ is

$$
\frac{1}{2}\ln\det(\Sigma_*) + \frac{d}{2}\ln(2\pi e),
\tag{20}
$$

Table 4: Comparison of training time and memory assumption across various graph self-supervised methods. For GRACE, the representation dimension is fixed to 256. MVGRL sets it to 256 on Pubmed and Computers. For other cases, the representation dimension is set to 512. "w/o org." means that the orthogonality between projection vectors is not required.

| Algorithm | Cora | | Citeseer | | Pubmed | | Computers | |
|---|---|---|---|---|---|---|---|---|
| | Time | Memory | Time | Memory | Time | Memory | Time | Memory |
| DGI | 6.8s | 3.8GB | 9.4s | 7.8GB | 44.9s | 11.2GB | 71.2s | 11.3GB |
| GRACE | 5.1s | 1.2GB | 7.4s | 1.5GB | 1,169s | 12.2GB | 362.8s | 7.4GB |
| MVGRL | 23.7s | 3.8GB | 48.4s | 7.9GB | 2,010s | 9.1GB | 78.8s | 16.6GB |
| ANA-IC w/o org. | 2.7s | 2.5GB | 2.0s | 2.6GB | 11.3s | 4.8GB | 5.3s | 3.9GB |
| ANA-IC | 3.8s | 2.5GB | 2.5s | 2.6GB | 12.3s | 4.8GB | 6.2s | 3.9GB |

where $\det(\cdot)$ denotes the determinant of a matrix. Assuming $\{\lambda_j | j = 1, 2, \ldots, d\}$ are $d$ eigenvalues of $\boldsymbol{\Sigma}_*$, $\det(\boldsymbol{\Sigma}_*) = \prod_{j=1}^{d} \lambda_j$ holds. Besides, $\sum_{j=1}^{d} \lambda_j = tr(\boldsymbol{\Sigma}_*) = d$. According to the AM-GM inequality (Hirschhorn, 2007), we can know that

$$
\begin{aligned}
\det(\boldsymbol{\Sigma}_*) \\
= \prod_{i=1}^{d} \lambda_i \\
\leq \left( \frac{\lambda_1 + \lambda_2 + \cdots \lambda_d}{d} \right)^d \\
= 1.
\end{aligned}
\tag{21}
$$

$\det(\boldsymbol{\Sigma}_*)$ achieves the upper bound of 1 when the eigenvalues $\{\lambda_1, \ldots, \lambda_d\}$ of $\boldsymbol{\Sigma}_*$ are all equal to 1, that is, the Gaussian entropy in Eq. (20) reaches the maximum value. According to Property 1, when the representations achieve isotropy, the eigenvalues of the covariance matrix will be equal, which satisfies the condition for the equality to hold in Inequality (21). Thus, we conclude the proof. □

# E   MORE EXPERIMENTS

## E.1   EFFICIENCY COMPARISON

**Efficiency Comparison.** To demonstrate the efficiency of our approach, we conduct a comparative analysis between it and other graph self-supervised learning methods, evaluating them based on two key factors: the time required for the training and the associated memory assumption. As depicted in Table 4, overall, our method demonstrates notable advantages, including shorter training duration and diminished memory usage in most instances. Not relying on projection heads, parameterized mutual information estimators, and negative samples greatly reduces memory consumption and computational burden. Besides, our method directly optimizes the representation space, allowing for quick convergence and shorter training time. When pairwise orthogonality between projection vectors is demanded, which involves performing a QR decomposition on a random matrix, it adds acceptable training time.

## E.2   VISUAL STUDIES

**Visualizations of t-SNE Embeddings.** To gain a deeper insight into our method, we provide a set of t-SNE (Van der Maaten & Hinton, 2008) plots depicting both the raw features and learned representations under various configurations in Figure 5. As shown in Figure 5(a), the 2-dimensional t-SNE embeddings of the raw features exhibit a high degree of overlap. The visualization in Figure 5(b), characterized by a messy elliptical shape, illustrates that the method without the ANA term can only learn diverse yet meaningless information. Figure 5(c) indicates that the method without the IC term can learn meaningful representations, as evidenced by discernible clustering in their 2-dimensional projections. However, it's worth noting that the two dimensions of t-SNE embeddings exhibit linear correlation (this phenomenon becomes more apparent after excluding the blue

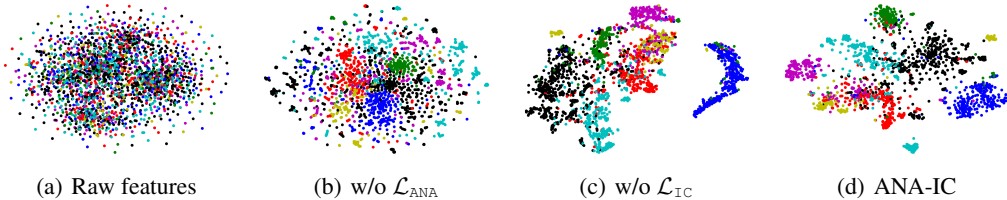

(a) Raw features      (b) w/o $\mathcal{L}_{\text{ANA}}$      (c) w/o $\mathcal{L}_{\text{IC}}$      (d) ANA-IC

Figure 5: t-SNE embeddings of the raw features and learned representations under various configurations on Cora dataset. "w/o" stands for "without". Best viewed in colors.

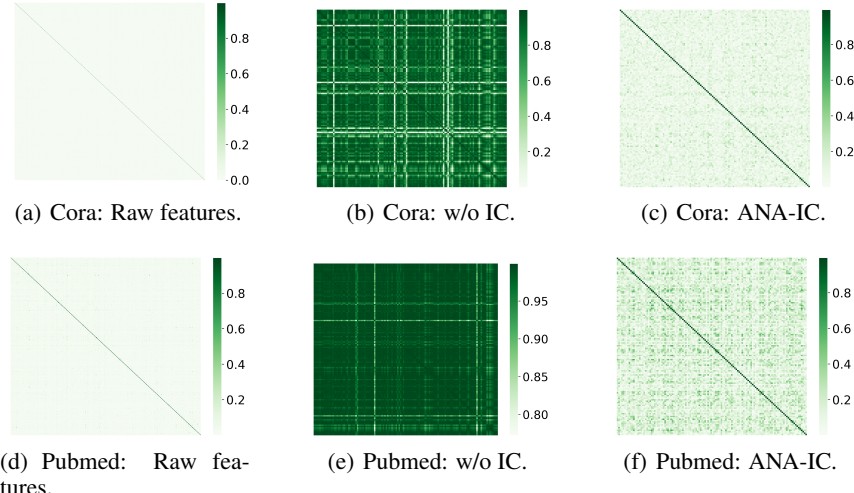

(a) Cora: Raw features.      (b) Cora: w/o IC.      (c) Cora: ANA-IC.

(d) Pubmed: Raw features.      (e) Pubmed: w/o IC.      (f) Pubmed: ANA-IC.

Figure 6: Visualizations of the correlation matrices with absolute values of raw features and representations under various configurations on Cora and Pubmed.

dots), which suggests that the dimensional collapse issue is potential. The visualization in Figure 5(d) demonstrates that our approach is capable of learning meaningful, diverse, and interpretable representations, which are better clustered based on their actual categories.

**Visualizations of Correlation Matrix.** To verify the effectiveness of the IC term in preventing dimensional collapse, we visualize correlation matrices of raw features, representations without IC term, and representations with IC term on Cora and Pubmed datasets, as shown in Figure 6. Concretely, for a matrix $\mathbf{Z} \in \mathbb{R}^{N \times D}$ characterizing node features or representations, which has been normalized to 0-mean and 1-standard-deviation, its correlation matrix is $\frac{1}{N}\mathbf{Z}^{\top}\mathbf{Z}$, where each element represents the Pearson correlation coefficient between two channels. For the convenience of visualization, we take the absolute value of each element in the correlation matrix. In Figure 6(a,d), the off-diagonal elements tend to be 0, which demonstrates that various dimensions of raw features of Cora and Pubmed have been well decorrelated. Various channels of representation matrix in Figure 6(b,e) are tightly coupled together with large off-diagonal elements, which suggests that dimensional collapse exists. The visualizations in Figure 6(c,f) demonstrate that our method, incorporating the IC term, is capable of effectively mitigating the dimensional collapse issue and learning highly disentangled and diverse representations.

**Toy Experiments for Isotropic Constraints.** We conduct some toy experiments to show the effects of isotropic constraints in preventing dimensional collapse and learning diverse representations. First, we build a simple neural network with three fully-connected layers. 4,000 data points are sampled from a Gaussian distribution, which construct a data matrix $\mathbf{x} \in \mathbb{R}^{4,000 \times 2}$ as shown in Figure 7(a). The input data points represent a strong correlation between the two dimensions and exhibit an anisotropic distribution. The outputs of the constructed simple model with randomly initialized parameters are shown in Figure 7(b), which still displays a typical anisotropic distribution and shows

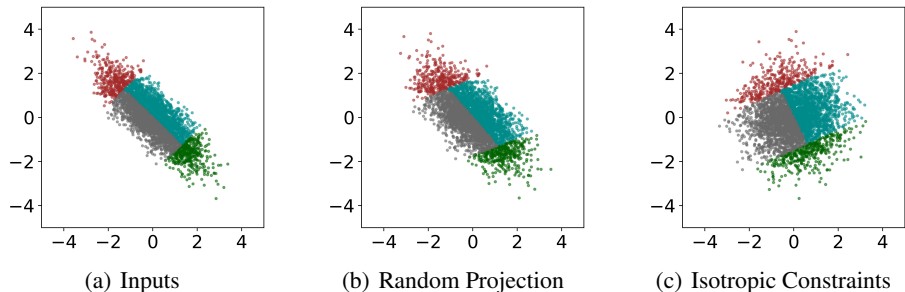

Figure 7: Visualizations of inputs and outputs of neural networks under various settings. For the convenience of visualization and comparison, the displayed points have been normalized to 0-mean and 1-variance. Colors are utilized to reflect the relative positions of points. Best viewed in colors.

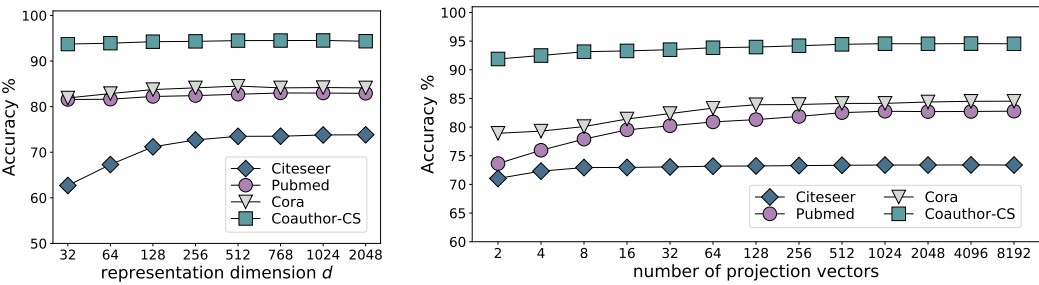

Figure 8: The effects of representation dimension.

Figure 9: The classification performance under various numbers of projection vectors.

a strong dependency relationship between the two dimensions. In Figure 7(c), after being trained with our isotropic constraints, the network can ultimately output decoupled representations, which exhibit isotropic appearance. The toy experiments demonstrate the effectiveness of our approach in mitigating dimensional collapse and enhancing representation diversity.

### E.3 HYPERPARAMETER SENSITIVITY ANALYSIS

**Effect of Representation Dimension.** We modify the representation dimension to investigate its impact on performance. Figure 8 presents the outcomes for different representation dimensions. It can be observed that classification accuracy shows an upward trend with increasing dimensions. Overall, the performance is not sensitive to dimension, which potentially demonstrates the robustness of our approach.

**Number of Augmented Views.** Generally, our method operates within a two-view framework. It is worth noting that our approach can readily be extended to a multi-view mode. Regarding the IC term, it is simple and straightforward to impose isotropic constraints individually on each view. The key challenge lies in how to achieve efficiently anchor-neighborhood alignment among multiple views. Taking three views as an explanation, their representations are $\mathbf{H}_A$, $\mathbf{H}_B$, and $\mathbf{H}_C$. Since each pair gives an upper bound in Eq. (6), we can get $\mathcal{L}_{\text{ANA}} = -\frac{2}{N} \cdot tr(\mathbf{H}_A^\top \mathbf{S} \mathbf{H}_B + \mathbf{H}_B^\top \mathbf{S} \mathbf{H}_A +$ $\mathbf{H}_A^\top \mathbf{S} \mathbf{H}_C + \mathbf{H}_C^\top \mathbf{S} \mathbf{H}_A + \mathbf{H}_B^\top \mathbf{S} \mathbf{H}_C + \mathbf{H}_C^\top \mathbf{S} \mathbf{H}_B) = -\frac{2}{N} \cdot tr(\mathbf{H}_A^\top \mathbf{S} (\mathbf{H}_B + \mathbf{H}_C) + \mathbf{H}_B^\top \mathbf{S} (\mathbf{H}_A + \mathbf{H}_C) +$ $\mathbf{H}_C^\top \mathbf{S} (\mathbf{H}_A + \mathbf{H}_B))$. Formally, it results in aligning one view with the sum (or average) of the other views. Assuming the number of views is $M$, the number of aligned pairs is $M$, not $2 \cdot C_M^2$. In other words, there is no need to enumerate each pair of augmented graphs. The relevant results are summarized in Table 5. With the increase of augmented views, there is a slight improvement in performance. However, the performance under two views is already quite substantial, which also possesses an advantage in efficiency.

Table 5: The node classification accuracy under various numbers of views.

| Number of views | 2 | 3 | 4 |
|---|---|---|---|
| Cora | 84.5 | 84.7 | 84.6 |
| Citeseer | 73.6 | 73.7 | 73.8 |
| Pubmed | 82.9 | 83.1 | 83.1 |
| Coauthor-CS | 94.54 | 94.63 | 94.65 |

Table 6: The node classification accuracy under various orders of central moments.

| Maximum order of moments | 2 | 3 | 4 |
|---|---|---|---|
| Cora | 84.5 | 84.6 | 84.6 |
| Citeseer | 73.6 | 73.6 | 73.6 |
| Pubmed | 82.9 | 83.0 | 83.1 |
| Coauthor-CS | 94.54 | 94.61 | 94.63 |

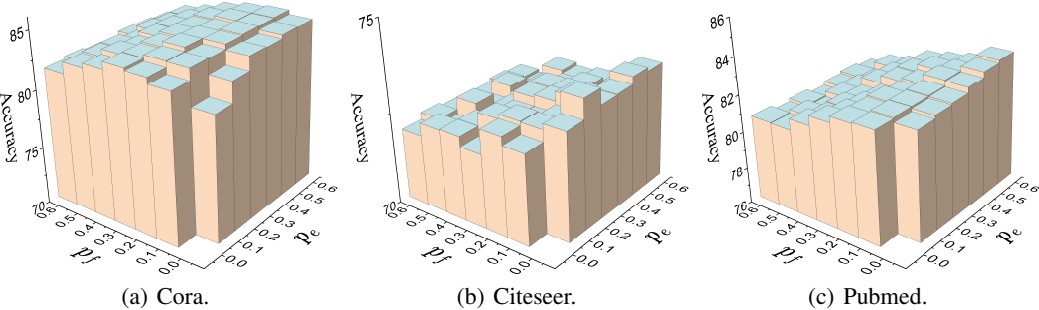

(a) Cora.       (b) Citeseer.       (c) Pubmed.

Figure 10: The classification performance under various combinations of feature masking ratio $p_f$ and edge removal ratio $p_e$.

**Number of Projection Vectors.** To enhance the diversity of node representations, we map representation points onto multiple projection vectors and expect to achieve isotropic distribution by reducing the distribution differences of projected points along distinct directions. Here, we investigate the influence of the number of projection vectors on node classification performance. As shown in Figure 9, the classification accuracy rigorously improves with the increase of the number of projection vectors. This result is unsurprising, as with an increasing number of projection vectors, they can more thoroughly permeate the entire representation space to achieve a better realization of isotropic distribution. It is worth noting that we do not emphasize the orthogonality between projection vectors in the experiments of Figure 9.

**Orders of Central Moment.** When imposing isotropic constraints, we employ central moments to describe the distribution of representations along a specific direction. Here, we investigate the impact of the maximum order of utilized central moments on performance. As shown in Figure 6, our approach can achieve satisfactory performance with only the 2-nd central moment. Adding higher orders of moments can further enhance performance. This is because adding orders of central moments allows for a more comprehensive description of the distribution of projections.

**Effect of Augmentation Intensity.** We conduct sensitivity analysis for augmentation intensity by examining the effects of various combinations of edge removal ratio $p_e$ and feature masking ratio $p_f$ on Cora, Citeseer, and Pubmed datasets. Overall, within an appropriate range of $p_e$ and $p_f$, our approach consistently achieves competitive results. Even when subjected to substantial augmentation, such as $p_e = 0.6$ and $p_f = 0.6$, our method continues to deliver satisfactory performance, reflecting the strong robustness of our approach.

### E.4 PERFORMANCE AND EFFICIENCY EVALUATION ON OGBN-ARXIV

Here, we evaluate the performance and efficiency of our approach on a large-scale graph Ogbn-Arxiv (Hu et al., 2020). The experimental results are summarized in Table 7, illustrating the effectiveness of our method. Besides, Figure 11 concurrently depicts the test accuracy and training time, which demonstrates that our approach can effectively reconcile both performance and efficiency simultaneously.

Table 7: Validation and test accuracy for Ogbn-Arxiv dataset. "OOM" means out-of-memory on a GPU with 24 GB of memory.

|  | Validation | Test |
|---|---|---|
| DGI | $71.19 \pm 0.24$ | $70.28 \pm 0.23$ |
| GRACE | $71.82 \pm 0.18$ | $70.91 \pm 0.21$ |
| GCA | $71.63 \pm 0.20$ | $70.77 \pm 0.22$ |
| GMI | OOM | OOM |
| MVGRL | OOM | OOM |
| BGRL | $72.58 \pm 0.14$ | $71.52 \pm 0.14$ |
| ANA-IC (ours) | $72.53 \pm 0.19$ | $71.49 \pm 0.17$ |

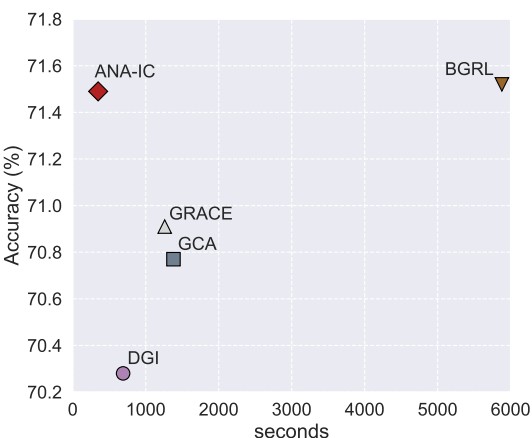

Figure 11: Test accuracy and training time on Ogbn-Arxiv.

## E.5 NODE CLUSTERING TASK

In this subsection, we evaluate the learned representations on node clustering task, compared to $k$-means, spectral clustering, DeepWalk (Perozzi et al., 2014), GAE (Kipf & Welling, 2016b), DGI (Veličković et al., 2018), MVGRL (Hassani & Khasahmadi, 2020), GRACE (Zhu et al., 2020), DNGR (Cao et al., 2016), RMSC (Xia et al., 2014), MGAE (Wang et al., 2017), and DAEGC (Wang et al., 2019). For the approaches not specifying the clustering algorithm, such as DeepWalk and ANA-IC, we apply the $k$-means algorithm on their learned representations. The clustering results are evaluated through three metrics: Normalized Mutual Information (NMI), Adjusted Rand Index (ARI), and F-score (F1). Because some baseline methods do not conduct clustering experiments or report incomplete indicators, we first learn node representations based on their official codes and then assess the learned representations under the same $k$-means algorithm for fair comparisons. For DeepWalk, the number of random walks is 10, the path length of each walk is 20, the context size is 10, and the number of embedding dimension is 256. For DNGR, the encoder is built with two 512-dim hidden layers and a 256-dim embedding layer. For RMSC, the trade-off parameter is set to 0.005. For MGAE, we set the number of layers, corruption level $p$, and regularization coefficient $\lambda$ to 3, 0.4, and $10^{-5}$ respectively. For DAEGC, the encoder is constructed with a 256-dim hidden layer and a 16-dim embedding layer, and the clustering coefficient is set to 10. The number of clusters is set to the number of ground-truth classes. Table 8 summarizes the experimental results. Our method showcases competitiveness across all datasets, suggesting the effectiveness of our method. Besides, our approach achieves outstanding performance in both node classification and node clustering tasks, demonstrating its strong generalization capability.

Table 8: Clustering performance in NMI, ARI, and F1. The best results are highlighted in boldface.

| Algorithm | Input | Cora | | | Citeseer | | | Pubmed | | |
|---|---|---|---|---|---|---|---|---|---|---|
| | | NMI | ARI | F1 | NMI | ARI | F1 | NMI | ARI | F1 |
| $k$-means | **X** | 0.377 | 0.149 | 0.415 | 0.241 | 0.154 | 0.399 | 0.263 | 0.237 | 0.542 |
| Spectral | **A** | 0.265 | 0.158 | 0.314 | 0.082 | 0.075 | 0.236 | 0.136 | 0.091 | 0.455 |
| DeepWalk | **A** | 0.401 | 0.254 | 0.541 | 0.238 | 0.085 | 0.372 | 0.251 | 0.203 | 0.605 |
| DNGR | **A** | 0.318 | 0.142 | 0.340 | 0.180 | 0.043 | 0.300 | 0.153 | 0.059 | 0.445 |
| RMSC | **X, A** | 0.320 | 0.203 | 0.347 | 0.308 | 0.266 | 0.404 | 0.273 | 0.247 | 0.521 |
| GAE | **X, A** | 0.397 | 0.293 | 0.415 | 0.174 | 0.141 | 0.297 | 0.249 | 0.246 | 0.511 |
| VGAE | **X, A** | 0.408 | 0.347 | 0.456 | 0.163 | 0.101 | 0.278 | 0.216 | 0.201 | 0.478 |
| MGAE | **X, A** | 0.489 | 0.436 | 0.531 | 0.416 | 0.425 | 0.526 | 0.271 | 0.224 | 0.634 |
| DAEGC | **X, A** | 0.528 | 0.496 | 0.682 | 0.397 | 0.410 | 0.636 | 0.266 | 0.278 | 0.659 |
| GRACE | **X, A** | 0.556 | 0.518 | 0.672 | 0.377 | 0.369 | 0.561 | 0.251 | 0.230 | 0.646 |
| DGI | **X, A** | 0.565 | 0.529 | 0.684 | 0.431 | 0.434 | 0.632 | 0.281 | 0.263 | 0.657 |
| MVGRL | **X, A** | 0.571 | **0.542** | 0.694 | 0.442 | 0.458 | 0.641 | 0.298 | 0.281 | 0.672 |
| ANA-IC (ours) | **X, A** | **0.582** | 0.531 | **0.701** | **0.447** | **0.462** | **0.647** | **0.332** | **0.331** | **0.693** |

---

**Algorithm 1** Overall Workflow of ANA-IC

**Input**: A graph $G(\mathbf{A}, \mathbf{X})$ with $N$ nodes, neural encoder $f_\theta$, balancing factor $\lambda$, augmentation function space $\mathcal{T}$, training epochs $T$, neighborhood size $K$, damping coefficient $t$, representation dimension $d$.

1: Initialize the neural encoder $f_\theta$;
2: Calculate affinity matrix $\mathbf{S}$ according to Eq. (3);
3: **repeat**
4:     Randomly sample two augmentation functions $\tau_A$ and $\tau_B$ from $\mathcal{T}$;
5:     Generate two augmented views $G'_A(\mathbf{A}'_A, \mathbf{X}'_A) = \tau_A(G)$ and $G'_B(\mathbf{A}'_B, \mathbf{X}'_B) = \tau_B(G)$;
6:     Obtain node representations $\widetilde{\mathbf{H}}_A = f_\theta(\mathbf{A}'_A, \mathbf{X}'_A)$ and $\widetilde{\mathbf{H}}_B = f_\theta(\mathbf{A}'_B, \mathbf{X}'_B)$;
7:     Get normalized representations $\mathbf{H}_A$ and $\mathbf{H}_B$;
8:     Generate two groups of $d$ orthogonal projection vectors $\{\mathbf{q}_j^A | j = 1, \ldots, d\}$ and $\{\mathbf{q}_j^B | j = 1, \ldots, d\}$;
9:     Calculate the loss $\mathcal{L}_{\text{ANA}}$ according to Eq. (6);
10:    Calculate the loss $\mathcal{L}_{\text{IC}}$ according to Eq. (10);
11:    Obtain the overall objective $\mathcal{L} = \mathcal{L}_{\text{ANA}} + \lambda \cdot \mathcal{L}_{\text{IC}}$;
12:    Update parameters $\theta$ through back propagation;
13: **until** reaching maximum training steps $T$
14: Get $\mathbf{H} = f_\theta(\mathbf{A}, \mathbf{X})$ for downstream tasks.

---

## F   PSEUDOCODE FOR OVERALL WORKFLOW

The overall workflow of ANA-IC is summarized in Algorithm 1. It is worth noting that the orthogonality between projection vectors is not enforced in the 8-th step, and the number of projection vectors is not necessarily constrained to the representation dimension $d$. The PyTorch-style code for isotropic constraints is presented in Algorithm 2.

---

**Algorithm 2** PyTorch-style code for Isotropic Constraints.

```python
# H: centrally normalized representations, shape=[N,d]
# n: number of projection vectors

def loss_ic_org(H):
    # Require orthogonality among projection vectors
    d = H.shape[1]
    random_mat = torch.randn(d, d)
    Q, _ = torch.qr(random_mat)
    H = H @ Q
    cm_2 = torch.var(H, dim=0)
    loss_ic = torch.var(cm_2)
    return loss_ic

def loss_ic_no_org(H, n):
    # Do not require orthogonality among projection vectors
    d = H.shape[1]
    random_mat = torch.randn(d, n)
    Q = random_mat / torch.norm(random_mat, dim=0, keepdim=True)
    H = H @ Q
    cm_2 = torch.var(H, dim=0)
    loss_ic = torch.var(cm_2)
    return loss_ic
```

---

