# OpenReview forum: "Boosting Self-Supervised Graph Representation Learning via Anchor-Neighborhood Alignment and Isotropic Constraints"
_ICLR.cc/2024/Conference — ICLR 2024 Conference Withdrawn Submission_

### Official Review · Reviewer_BwDP · 2023-10-30

**Soundness:** 1 poor
**Presentation:** 2 fair
**Contribution:** 1 poor
**Rating:** 3
**Confidence:** 4

**Summary:**

This paper first revisits several remaining issues limit the capability of existing graph self-supervised methods: Then, it proposes two complementary components to address these issues. The first is Anchor-neighborhood alignment strategy, which uses graph diffusion to construct a probability distribution of positive samples based on the structural context of the anchor node. The other is Isotropic constraint strategy, which encourages node representations to exhibit a consistent distribution in space, promoting diversity in representations. Finally, it conducts some experiments to evaluate the proposed method, showing that the proposed method outperforms both contrastive and generative state-of-the-art baselines on several tasks across multiple datasets.

**Strengths:**

1.	It provides some theorical support for the proposed model.
2.	It tests on several widely-used datasets, and the proposed method can sometimes beat the existing methods.
3.	The authors provide their codes, although it is hard to reproduce the reported results.

**Weaknesses:**

1.	The reported results cannot be reproduced, with the given codes and suggested hyper-parameters. I have re-run the code on three famous citation networks, and only get {cora: 83.9+/- 0.76, citeseer: 72.94+/- 0.72, and pubmed: 78.67+/- 0.45}. The reproduce dependencies are Python 3.11.5, PyTorch 2.0.1, DGL 1.1.2 and scikit-learn 1.3.1.
2.	The dataset split is not the standard. It is well known that the DGL’s dataset split is different from the classical GCN work. Therefore, the experiments can be improved.
3.	Some theories, that not original, should not show in this paper. For example, Lemma 1 Lemma 2, and Property 1 are classical ones. You can easily find them from a text book.
4.	Confusion of symbol system. For example, the loss symbol in Eqs. 5 and 6 are quite similar.
5.	Some refences are missing. For example, Isotropic Constraints (IC).
6.	The work could be largely improved. The proposed method is very complicated and also contains lots of hyper-parameters, which makes it hard to reproduce.
7.	The hyper-parameter setting is wired. As we can see from the readme file in the code Repository, the hyper-parameters are kind of very fine-grained, like pe (0.35, 0.45, 0.6) and lambd (30, 100, 280). It is well known that tuning on the valuation set is not easy for such fine-grained search space. Or these ones are tunned to choose the best results on the test set?

**Questions:**

1.	Why the reported results cannot be reproduced?
2.	Which theories are original of this paper?
3.	Is it useful to design such a complex model?
4.	The hyper-parameters are chosen based on the test set?
5.	See the weakness in the “*Weaknesses” part.

---

### Official Review · Reviewer_MQvP · 2023-10-30

**Soundness:** 2 fair
**Presentation:** 2 fair
**Contribution:** 2 fair
**Rating:** 5
**Confidence:** 4

**Summary:**

In this paper, the authors propose to find anchors in one augmentation view as the positive pairs of points from another augmentation view for graph contrastive learning. The contrastive loss is different from the classical InfoNCE loss. To prevent the underlying dimensional collapse, the authors also propose isotropic constraint, which is based on high-order central moment on random $n$ directions.

**Strengths:**

- The idea of using anchors, instead of the same point from different augmentation views, seems rational and promising.
- The experimental discussions seem sufficient.
- The paper is easy to follow and understand.

**Weaknesses:**

- The motivation of using anchors is somewhat unconvincing. In Page-4, the authors claim:

  ```
  Disregarding the potential role of the topological structure in shaping self-supervised objectives and treating it solely as a regulator of message passing in GNNs lead to inadequate utilization of graph information.
  ```

  Since $h_v^*$ has implicitly contained the structural information as long as $h_v^*$ is represented by GNNs,  why should the graph structure be explicitly considered in contrastive losses? By the way, what is the exact meaning of regulator?

- I find a paper (AnchorGAE: General Data Clustering via O(n) Bipartite Graph Convolution) that also applies anchors to GNNs but the authors seems not to discuss the relation between this paper and it. It seems that Section 3.2 is somewhat similar to the above paper.

- Although the concern of dimensional collapse is rational, the conducted ablation experiments of IC (Table 2, Figures 5-6) seems not to show  the apparent impact of dimensional collapse. It is hard to say that the IC module really works.

- From the conducted experiments, it is hard to verify whether the contrastive learning with anchors outperforms the classical scheme. It seems that the results achieved by the proposed method without $\mathcal{L}_{\rm IC}$ is unsatisfactory.

- The writing quality can be further improved. For example,

  -  In eqn (3), an N-dimension vector $r$  is denoted by $\mathbb R^{N \times 1}$ while a d-dimension vector is written as $\mathbb R^d$ in eqn (7).
  - In Figure 2, the meaning of the middle graph (with green background) is unclear. Should it be bipartite between any two augmented views?

**Questions:**

Please see Weakness.

---

### Official Review · Reviewer_CqLJ · 2023-11-10

**Soundness:** 2 fair
**Presentation:** 2 fair
**Contribution:** 2 fair
**Rating:** 5
**Confidence:** 3

**Summary:**

This paper introduces a novel graph self-supervised learning approach aimed at addressing the well-known issue of dimensional collapse within the self-supervised learning paradigm. Specifically, the paper puts forward an anchor-neighborhood alignment strategy accompanied by an isotropic constraint. The effectiveness of the proposed ANA-IC method is extensively assessed, revealing that it yields superior performance compared to existing methodologies.

While the success of the proposed method in outperforming existing Graph Neural Networks (GNNs) is promising, the specific reasons why anchor-neighborhood alignment effectively resolves the problem remain somewhat ambiguous.

**Strengths:**

1. The proposed method outperforms existing GNN methods.

**Weaknesses:**

1. The rationale behind introducing anchor-neighborhood alignment lacks clarity. Although the introduction asserts that the proposed approach aims to circumvent dimensional collapse in self-supervised learning, the underlying logic remains somewhat ambiguous.
2. The comparison conducted by the proposed method primarily involves methodologies from the year 2022. Given the rapid pace of advancements in the field of machine learning, it would be beneficial for the paper to include a comparison with the most recent state-of-the-art methods.

**Questions:**

1. As highlighted in the weakness section, the rationale behind the introduction of anchor-neighborhood alignment requires a more comprehensive explanation. It is imperative to elucidate precisely how the utilization of anchor-neighboring alignment addresses the issue of dimensional collapse in self-supervised learning.
2. The current evaluation of the proposed method is primarily based on empirical comparisons with baseline methodologies established in 2022. To enhance the robustness of the evaluation, it is crucial to incorporate additional comparisons with recently proposed methods from 2023. Furthermore, it is essential to articulate the specific advantages that the proposed method offers over these contemporary approaches to highlight its superiority and unique contributions to the field.